# Vehicle Bump Testing Parameters Influencing Modal Identification of Long-Span Segmental Prestressed Concrete Bridges

**DOI:** 10.3390/s22031219

**Published:** 2022-02-05

**Authors:** Wilson Hernandez, Alvaro Viviescas, Carlos Alberto Riveros-Jerez

**Affiliations:** 1School of Civil Engineering, Industrial University of Santander, Carrera 27 Calle 9, Bucaramanga 680002, Colombia; wilson2198182@correo.uis.edu.co (W.H.); alvivija@uis.edu.co (A.V.); 2Faculty of Engineering, University of Antioquia, Calle 67 No. 53-108, Medellín 050010, Colombia

**Keywords:** prestressed concrete box girder bridge, ambient vibration test, forced vibration test, vehicle-induced vibration test, modal identification, optimal sensor placement, numerical simulation

## Abstract

In-service prestressed concrete box girder bridges have received increasing attention in recent years due to a large number of bridges reaching decades in service. Therefore, the ageing of infrastructure demands the development of robust condition assessment methodologies based on affordable technology such as vehicle-induced vibration tests (VITs) in contrast with more expensive existing technologies such as tests using hammers or shakers. Ambient vibration tests (AVTs) have been widely used worldwide, taking advantage of freely available ambient excitation sources. However, the literature has commonly reported insufficient input energy to excite the structure to obtain satisfactory modal identification results, especially in long-span concrete bridges. On the other hand, the use of forced vibration tests (FVTs) requires more economic resources. This paper presents the results of field measurements at optimally selected locations in VITs consisting of a 32-ton truck and a springboard with a height of 50 mm. AVTs using optimal sensor placement (OSP) provide similar results to VITs without considering OSP locations. Additionally, the VIT/AVT cost ratio is reduced to 2 since a shorter data collection time is achieved within a one-day (8 h) test framework, which minimizes temperature effects, thus leading to improvements in AVT identification results, especially in vertical modes.

## 1. Introduction

Existing prestressed concrete (PC) segmentally constructed bridges have received increasing attention in recent years due to a considerable number of in-service bridges that have suffered from excessive mid-span deflection, thus restricting the construction of PC bridges with larger spans [1]. Therefore, the ageing of in-service bridges and the construction of bridges with longer spans demand the development of non-destructive test (NDT) methods for structural condition assessment to ensure safe operation by considering structural degradation and environmental factors. Several structural health monitoring (SHM) approaches have been developed over the past two decades based on artificial intelligence, big data, statistical pattern recognition, and the combined use of accelerometers and cameras [2,3,4,5,6]. The variation of modal parameters is related to changes in physical properties of a bridge, and field studies have shown that environmental and operational factors, such as temperature, traffic, humidity, and solar radiation, induce changes in modal parameters commonly used in vibration-based damage detection methods, temperature variation is the critical source of modal variability. Changes in modal frequencies caused by temperature variation may reach 10%, showing values higher than those related to changes in modal frequencies due to structural degradation. Structural damage identification in practical applications using vibration-based data is still challenging because the process of structural degradation is accumulated over a long time scale [2,5].

Continuous dynamic monitoring of the Westend bridge, a prestressed concrete bridge, was conducted using data acquired from 2000 to 2013 [5]. A significant influence of environmental and operational effects in modal identification was mainly associated with temperature variation. Temperature variations influenced the natural frequencies of the first five modes in the range of 7.4% to 17.5%, showing gradual decrements in the 14-year study. In addition, the loss of prestress in the tendons was a source of natural frequencies decrement. The facts mentioned above show that continuous dynamic monitoring requires a quantitative understanding of the temperature effect and how this factor influences the modal information collected at different time scales. Ye et al. [1] successfully separate the effects of live load, temperature, and structural deflection from deflection signals obtained from numerical models up to noise levels of 10%, field validation was greatly affected by environmental factors. Analysis of data collected during one year from a long-span PC bridge with a main span of 160 m (Hanxi bridge) showed that structural deflection caused by structural damage and material deterioration was possible to extract. However, other effects related to drifting, bias, precision degradation, and gain significantly affect structural condition assessment in field applications. Ye et al. [1] concluded that the Hanxi bridge had suffered from an incremental deflection by only using deflection signals, but the monitoring system deployed in the Hanxi bridge also consists of accelerometers. In this framework, more robust monitoring systems are necessary, and therefore deflection measurements can significantly benefit from dynamic assessments.

It is essential to consider the relevance of bridge modelling to study factors, such as live load, temperature, concrete shrinkage, and creep, and prestress loss that could affect data collected from sensors in real applications [1]. Therefore, a robust SHM approach must consider structural and operational and environmental factors. Thus, different sensor technologies and algorithms are envisioned to address the effect of simultaneous actions. Field data results have proven to be more successful when complemented with finite element (FE) modelling. AVTs have been widely used worldwide, taking advantage of freely available ambient excitation sources, such as wind and traffic loads. The scientific literature has commonly reported the lack of sufficient input energy to excite structural modes of interest, and a longer sensing duration requires careful consideration in selecting the sensing system related to the number of sensors, power, board memory, and data transmission [7].

On the other hand, the use of FVTs requires more resources becoming in some contexts an unfeasible option, especially for long-span PC bridges due to the level of forces needed to excite such bridges significantly. Emerging wireless sensor technologies still need to address existing limitations, such as drift phenomenon, communication bandwidth, data loss, time synchronization, and signal length. Therefore, conventional wired systems are still widely used [7]. In this framework, VIT is becoming an attractive option to overcome the limitations of AVT and FVT because field implementation is a relatively straightforward process. However, such deployments are currently restricted to small- and medium-span bridge structures. Due to limitations in estimating the excitation force exerted by a moving vehicle, VIT is not classified as an FVT.

Tan et al. [8] studied the dynamic response of simply supported bridges using vehicle bump testing by dividing the excitation process into four stages, namely static condition before the test, vehicle bump process, and free-decay response after vehicle bump, respectively. The numerical validation showed that the initial condition given by the second stage is crucial and greatly affects the relationship between bump height and bending moment. Tan et al. [8] defined vehicle bump testing as a unique vehicle-bridge coupled process where the rear wheels are always in contact with the bridge deck. On the other hand, front wheels are in contact with the bridge deck, but before falling, separate from the bridge deck. A three-axle 34-ton truck and a springboard with a height of 150 mm, placed at midspan, were used for vehicle bump test validation in the Julongue bridge, a simply supported bridge with a span of 16 m. Although field data collected from strain sensors strategically located on the bridge allowed the identification of similar trends in strain time-history curves obtained from numerical analysis and field data, maximum strain values showed errors of approximately 12%. Tan et al. [8] concluded that the three main factors related to errors are using a three-axle vehicle in contrast with the numerical derivation of the proposed procedure using a two-axle vehicle, the existence of reinforced concrete pavement that affects the transverse distribution of the vehicle load, and environmental factors. It is essential to highlight that, in field testing, the truck moves to the top of the springboard very slowly and brakes before putting the front wheels above the springboard. Therefore, braking forces are negligible. Although [8] using numerical simulation defined a limit bump height of 230 mm, this parameter remains crucial in field applications and the influence of environmental factors in the response of the bridge.

Zhang et al. [9] proposed an additional virtual mass method for damage identification using vehicle bump-induced excitation defined by the authors as an impact applied to the bridge. Acceleration records were collected from 10 sensors equally spaced in a numerical model of a two-span bridge with equal span lengths of 25 m. The first three natural frequencies and corresponding mode shapes were used to conduct damage identification. Although the 10-sensor configuration successfully identified damage, the four-sensor configuration (quarter-span sensors) showed damage prediction errors up to 20.8%. Zhang et al. [9] concluded that the proposed method requires a considerable number of sensors if modal identification is used to implement the method. An essential factor to be considered in the field validation of the method is using a reduced number of sensors. Difficulties in practical applications also arise from challenging factors related to the unknown nature of structural parameters, such as mass, stiffness, and damping, which are needed to model the structure. Therefore, using dynamic parameters such as frequencies and mode shapes obtained from field measurements greatly benefit the above-mentioned theoretical developments, primarily when structural degradation is the main objective. On the other hand, vehicle characterization in terms of mass and stiffness becomes challenging due to the complex nature of the vehicle system. Although the structural damage assessment of bridges using vehicle bump testing based on numerical simulation has shown that structural damage is detected even in the presence of high noise levels, as previously mentioned, field validation remains critical. In addition, the limited number of instrumented nodes impose difficulties in obtaining accurate modal identification results. The implementation of the procedures mentioned above in long-span PC bridges needs a preliminary analysis concerning changes in the geometry of the cross-section of the deck, which is sequentially constructed. Thus, stiffness cannot be assumed constant. In addition, vehicle loads applied at a specific section of a long-span PC bridge might not provide sufficient impact energy to excite the structure. Therefore, it is crucial to identify the most influential parameters related to vehicle loads in long-span PC bridges and use field data to validate numerical modelling of such parameters.

As previously explained, in vehicle bump testing, the truck’s movement is restricted to generating the impact force after passing the road bump. Although it is currently possible to model the bridge response using bump-induced excitation in regular simply supported and two-span bridges, the number of required parameters makes it not convenient for practical application in long-span PC bridges, where the lack of uniformity of a cross-section of long-span PC is the main limitation. However, an alternative method for practical implementation in long-span PC bridges can be adapted if the truck reduces the speed before it reaches the road bump and just after passing over a road bump generates an impact force on the bridge deck. The simplest way to conceptualize this phenomenon corresponds to a parabolic projectile motion with known speed and height as a function of launch angle and initial velocity. However, when the truck’s axle falls on the bridge deck, a rebound occurs, implying a dissipation of energy by an inelastic collision related to an unknown coefficient of restitution, which depends on the mechanical properties of the truck suspension. In this framework, the impact force involves variables that are difficult to characterize in the field, such as tire pressure.

Therefore, a more practical procedure such as the one presented by [10] is adopted in this paper. Gatti [10] employed a two-axle 2-ton truck moving at a speed of 30 km/h on a three-span simply-supported concrete bridge to generate impact loads following a vehicle jump obtained with a 100 mm high springboard. A static load test was also conducted and compared with the results of the vehicle load test. Gatti [10] showed that identification of the dynamic properties of the bridge was only possible using data collected from vehicle load test and recommends the use of dynamic loads to test post-construction bridges based on the fact that structural degradation leading to stiffness loss can be identified using vibration data. In addition, bridge deflection can be calculated from an updated FE model using collected vibration data. In Colombia, the current bridge design code [11] does not indicate the procedure for dynamic testing of new and existing bridges. In addition, long-span PC bridges have become widely used across the country for highway construction. Therefore, in order to provide basic guidelines for the characterization of segmental bridges using dynamic testing procedures and based on the studies mentioned above [1,7,8,9,10], this paper presents the analysis of field measurements collected from a long-span PC segmentally constructed bridge located in Colombia. A 32-ton truck was used in conjunction with a springboard height of 50 mm to excite the bridge. The main objective is to propose a one-day test (8 h) to minimize the effect of temperature in modal identification. A FE model is used to implement an OSP approach known as effective independence (EI). As highlighted by [12], abundant numerical and purely experimental data have been provided, but more work related to practical applications using physics-based models are needed. Therefore, this paper presents the results of OSP measurements collected from field tests using vehicle-induced excitations to show the advantage of using optimally selected locations in vehicle bump testing in a test conducted during daylight (8 h) to reduce costs.

## 2. Study Case: The Tablazo Bridge

The Tablazo bridge was built to provide a road connection between Bucaramanga and San Vicente de Chucurí (Santander, Colombia), as shown in Figure 1. It is a long-span PC segmentally constructed bridge opened to traffic in 2014 with span arrangements of 93 + 183 + 183 + 93 m and a constant width of 10.55 m (Figure 2). Its columns range in height from 57 m to 109 m, and the minimum distance between the bottom of the bridge deck and the water level is 32 m. The bridge consists of 115 cast-in-place concrete segments whose height varies from 9 m on the piers to 3 m at mid-span. ASTM A416, Grade 270 was used to define the material properties for the prestressing tendons with ultimate stress of prestressing strands fu = 1860 MPa. 12-wire and 19-wire strands with a nominal diameter of 5/8” are employed as prestressing tendons. The curvature friction coefficient μ and the wobble friction coefficient k are 0.20 and 0.0016/m, respectively. Gatti [10] used preliminary FE analysis to identify the structure’s first vibration mode, and then the dynamic test was executed in a three-span PC bridge with approximately equal span lengths of 18 m. Although [10] defined a seismic load (dynamic tremor alpha = 0, which represents a vertical action proportional to the value of the vehicle weight) to identify the first natural frequency of the bridge before field testing, in OSP implementations, the number of modes shapes becomes a crucial factor. Therefore, the FE model developed for the Tablazo bridge is dynamically tested using vehicle loads to determine the number of modes effectively identified under such load scenarios. In this framework, based on the construction drawings and specifications provided by the Government of Santander, a three-dimensional FE model is developed for the Tablazo bridge using MIDAS Civil©, as shown in Figure 3. Concrete compressive strength of 35 MPa for girders and piers is used, and a total of 1380 beam elements discretized every 0.5 m are considered to model the bridge. Fixed support at the base of piers is assumed but without considering any moments developed in the span connection to bridge abutments. Modelling of prestressing tendons is also included in the FE model. Table 1 shows the numerical mode shapes with frequencies ranging from 0 to 4 Hz.

## 3. Preliminary Analysis for Influential Parameters

The development of the field test proposal is based on the schematic flow chart shown in Figure 4, starting from conceptualization to identify the main influential parameters and then study their relevance using FE modelling. It is important to note that the main objective is to optimize the economic resources needed to conduct field testing and the costs associated with the bridge’s closure. AVTs are conducted first to identify OPS locations previously determined using the FE model. VITs are then conducted using OSP locations to extract modal information for FE model updating. A review of the scientific literature related to the dynamic characterization of bridges under different excitation techniques allows the identification of the most influential parameters [13,14,15,16,17,18,19,20,21,22], which are bridge typology, main span length (L), excitation source, sensor-to-sensor separation distance, time-series length, and sampling frequency. Figure 5 shows the distribution of the variables as a function of the main span length L. It is possible to observe that the excitation sources most widely used are ambient and vehicles, the sensor separation distance mainly ranges from L/4 to L/9, the sampling frequency ranges from 100 and 256 Hz, and the time-series length ranges from 1.5 and 15 min. Considering the importance of the time-series length as highlighted by [23,24], two time-series lengths are selected in the present study. The first value corresponds to 20 min, for sensor-to-sensor separation distance of L/10 (18 m), and the second value corresponds to 35 min, for sensor-to-sensor separation distance of L/6 (30 m).

It is possible to define the effect of a moving load (input force) acting on a bridge as a transitory test. Although input forces associated with moving loads cannot be measured in practical applications, in such loads, parameters such as speed and load weight can be controlled and applied at will, allowing improved identification of the dynamic properties [25]. Such moving loads, defined as vehicle loads, are associated with a complex phenomenon involving variables, such as vehicle weight, axle distribution (quantity-spacing), driving speed, mechanical properties of the suspension, pavement roughness, and vehicle-bridge interaction (VBI). These variables, typical of the excitation source, have been previously studied by [26,27]. VBI is related to the frequency coupling between the vehicle and the bridge, leading to frequency changes as a function of the mass ratio between the bridge and the vehicle. Pavement roughness should not be omitted since the thickness of the layer is related to the level of vibrations generated [28]. A study conducted by [29] identified that vehicles of equal weight but with different suspension properties altered the frequencies by increasing or decreasing the values concerning the frequencies obtained from AVTs. Numerical simulations are conducted to study the relevance of modal testing parameters in the dynamic characterization of long-span PC bridges and analyze the suitable configurations for vehicle loading in the context of field applications. It is important to note that factors, such as vehicle mechanical suspension properties, pavement roughness, and tire pressure, are out of the scope of this work due to practical limitations related to the precise determination of those parameters. However, vehicle-related parameters, such as vehicle weight, axle arrangement, axle spacing, vehicle speed, impact, and braking forces are considered in FE simulations. It was found in the literature review that trucks weighing more than 20 tons and travelling at speeds of between 10 and 80 km/h are widely used in field testing [10,27,30,31,32,33,34].

Based on the parameters mentioned above, time history analysis is conducted using a three-axle truck with a weight of 32 ton travelling at speeds of 10, 20, 40, and 60 km/h. Three load cases are selected. The first case corresponds to one truck passing over the bridge at constant travel speed, the truck passing over the bridge 2 and 3 times at constant travel speed and a combination of travel speeds. The design speed of the access road to the bridge is 60 km/h. The following load cases are considered for 2 repetitions and travel speed combinations. The repetitions were defined as 2 and 3 times at travel speeds of 20, 40, and 60 km/h and travel speed combinations (km/h) defined as C1:20–40, C2: 40–60, C3:20–20–40, C4:40–40–60, C5:20–40–40, C6:20–40–60, C7:40–60–60, C8:20–60, C9:20–20–60, C10:20–60–60. The distribution of sensor-to-sensor separation distance between L/2 (93 m) to L/10 (18 m) was selected, which implies the use of 8–30 equidistant recording nodes in the FE model distributed from 4–15 sensor configurations (movable sensor configurations) in sets of 3 (two movable sensors, one reference sensor), in analogy to running a full-scale dynamic test. The sampling frequency and time-series length were defined as 200 Hz and 200 s, respectively. Since the input signal is in this vertical direction, the only mode shapes identified in the numerical simulations are the 1-X (f = 0.492 Hz), 1-Z (f = 0.841 Hz) and 1-RY (f = 1.742 Hz) modes. Simulation results were verified using the modal assurance criterion (MAC) index, percentage of error in frequency and magnitude of the peak in the power spectrum with greater modal adjustment. MAC varies from 0 and 1, with 1 indicating a completely consistent mode shape. The acceleration responses for the nodes of interest were processed in ARTeMIS using the frequency domain decomposition (FDD) technique over a frequency range of 0–10 Hz with a frame of 1024 data points and a Hanning window of 66%. In order to select the frequency and the associated modal shape, the afferent width of 0.1 Hz was taken to the left and right of the highest magnitude peak in the power spectrum to select the frequency with the highest MAC value in each mode. The results of the MAC index are shown in Figure 6. The best modal consistencies are obtained for sensor-to-sensor separation distances of L/2 and L/3, followed by L/10. It is important to note that as repetitions and travel speed combinations increase, modal identification improves. Since the L/6 distribution contains L/2 and L/3 distributions, it is feasible to use it for field applications. As modal consistency improves by increasing repetitions and travel speed combinations, the relative error percentage decreases, being less in sensor-to-sensor separation distances of L/2 and L/3, as shown in Figure 7.

Figure 8 shows that the magnitude of the power spectrum decreases from L/2 to L/6 but increases from L/7 to L/10. It is essential to highlight that the frequency peaks may vary in field applications as a function of the input signal, but the distribution of the “energy content” of the signal as a function of the sensor-to-sensor separation distances is an important indicator. Figure 6, Figure 7 and Figure 8 show that the best dynamic characterization results for the selected mode shapes were obtained for sensor-to-sensor separation distances of L/6 (implicitly L/2–L/3) and L/10. These results are consistent with the review of the previous field experiments. The travel speed combinations for sensor-to-sensor separation distances of L/6 and L/10 are presented in Figure 9, Figure 10 and Figure 11. It is possible to observe that the combinations C9 and C10 show the best modal adjustment and the lowest frequency error but the lowest energy peak magnitudes in the spectrum power. However, taking into account the uncertainty in the field and the multiple input sources that could attenuate the vehicle’s signal when driving, the combination C2: 40–60 is selected for sensor-to-sensor separation distances of L/6 and L/10 because it shows the best modal adjustment, a lower percentage of frequency error, and magnitude of frequency peaks higher than the remaining combinations. Although the above combinations are results of numerical procedures allowing a preliminary identification of the suitable values to be used in field testing, deviations in full-scale tests may occur due to the ability of the truck driver and other external factors beyond the control of the authors (mechanical properties of the truck suspension, pavement roughness, and environmental factors).

## 4. Ambient Vibration Tests

Two high-sensitivity accelerometers Obsidian Kinemetrics^®^ were used to acquire and process data from the vibration signals (vertical (z) channel 1, longitudinal (x) channel 2, transverse (y) channel 3). The first sensor was located in a fixed position (reference sensor) at a distance of 390 m measured from the north access. The second sensor was moved sequentially from the north to south access in the positions shown in Figure 12 and Figure 13. At each new position of the movable sensor, data was recorded in time intervals of 35 min for the L/6 sensor configuration, and 20 min for L/10 sensor configuration. Hence, four AVTs were carried out on the Tablazo bridge. AVT1 and AVT2 (conventional) used sensor locations selected from the literature review in conjunction with the sensitivity analysis previously performed. AVT3 and AVT4 (hybrid) combine OSP with conventional locations. OSP locations were determined using an algorithm developed by [35]. OSP is based on the determination of available sensors and the modal shapes of interest. It is essential to highlight that the objective of this research is the optimization of modal identification results. Therefore, the EI method is selected in this study due to its better performance in practical applications [22]. The EI method was developed by [36] and selects optimal locations by identifying the best set of sensor locations from all possible positions, guaranteeing the linear independence of the mode shapes. Basically, from information obtained from numerical simulations, the mode shapes with the most significant possibility of being obtained in the field are selected to identify the coordinates of maximum strain energy or curvature changes. Based on the data collected from AVT1 and AVT2, the 11 identified modal forms are presented in Table 1. Figure 12 shows the location of sensors for the AVTs and the water level measured on the days that the dynamic tests were conducted.

The height of the submerged column is defined as the environmental factor responsible for water–structure interaction and, therefore, will be considered when updating the FE model of the bridge. The circled dots (AVT1 and AVT2) correspond to sensor locations at L/10 and L/6, respectively. AVT3 and AVT4 use squares to show OSP locations, also showing in circled dots conventional locations. AVT3 and AVT4 use locations of L/6 and L/10, respectively. The time-series length by location in AVT1 and AVT4 was 20 min, and AVT2 and AVT3 was 35 min. The objective is to conduct one test per day. Vertical mode shapes were best identified by both hybrid configurations (AVT2 and AVT3). Therefore, AVT2 and AVT3 are selected for vehicle-induced excitation tests (VITs), considering a reduced number of sensor locations.

## 5. Vehicle-Induced Excitation Tests

A three-axle 32-ton truck was used with a springboard height of 50 mm (Figure 14). Based on previously discussed field results provided by [8,10], the springboard was located in a pier support and midspan of the bridge deck, defined as Road Bump in locations 3 and 12 as shown in Figure 13. The tests required the bridge to be closed to vehicular traffic. Hence, two VIT (VIT1 and VIT2) tests were conducted using the sensor distribution of each L/6 (AVT2) and L/6 with OSP locations (AVT3). A stopping braking distance of 50 m was selected to ensure that the truck is indeed capable of stopping within the required distance, and therefore, those locations are defined as B (truck moving forward) and S-B (truck moving in reverse) (Figure 13). The previously selected load case C2 (40–60 km/h) is implemented in field testing, then VIT1 is defined when the truck starts moving with a constant speed of 40 km/h over the bridge. Braking force is then applied from locations 9–12 before the truck jump and braking force is again applied in 17 to stop the truck. Finally, the truck moves in reverse with a constant speed of 20 km/h to the starting point but applies braking forces in locations S-B. VIT2 is defined as similar to VIT1, but with a constant forward speed of 55 km/h. The time-series length per location is set to 10 min.

## 6. Comparisons

The acceleration responses were processed in ARTeMIS using the following techniques: FDD, enhanced frequency domain decomposition (EFDD), and stochastic subspace identification (SSI). The frequency range adopted was from 0 to 5 Hz with a frame of 1024 data points and a Hanning window of 66%. For the FDD technique, the afferent width of the peak with the highest energy per 0.5 Hz mode is used to select the frequency with the best MAC modal adjustment with respect to the FE model. For the EFDD technique, frequencies with a MAC rejection level greater than 0.9 are selected. For the SSI technique, 100 eigenvalues are defined as the maximum dimension with maximum frequency deviation and MAC values of 0.002 and 0.05, respectively, and a maximum coefficient of variation is defined as 0.1 in frequency. It is essential to mention that, depending on the data processing technique, there are differences in the results that will be further analyzed. A preliminary analysis of the modal identification results shows that VITs provide better results for mode shapes associated with frequencies higher than 1.8 Hz (Figure 15). To further analyze the dynamic results of the six tests performed on the Tablazo bridge, six study parameters are defined, comparing the average values of MAC indices of the 11 identified mode shapes (Table 2). The six parameters are defined as:The incidence of sensor location in AVTs with full time-series length.The incidence of time-series length in AVTs by equaling time-series length to 20 min.The incidence of excitation source between AVTs and VITs with full time-series length.The incidence of OSP in VITs with respect to AVTs with full time-series length.The incidence of AVTs with time-series length equals to VITs (10 min).The cost comparison between both types of dynamic tests.

### 6.1. FE Model Updating

In a preliminary analysis of the experimental frequencies (Table 3) with respect to the frequencies obtained from the FE model (Table 1), it is possible to observe differences up to 15% (average: 3%). This preliminary analysis shows the incidence of environmental factors to be considered when updating the FE model. Environmental factors, such as temperature, humidity, and surrounding water in submerged columns can induce variations in dynamic properties as reported by [25,26]. The water–structure interaction effect is defined by [37,38] as the effect of hydrodynamic pressure represented by a mass added to the structure as a function of the surrounding water mass. The added mass approach proposed by [39] is used to consider the phenomenon of water-structure interaction by adding the masses calculated according to equation C.3.7.3.1-1 AASHTO LRFD Bridge Design Specifications [40]. The water level heights recorded in the six dynamic tests are assigned to the corresponding nodes of the FE model, the mass value is calculated using (1) [40], where $C_{D}$ is the dimensionless drag coefficient, *w* is the specific weight of water $\left(N/m^{3}\right)$, *V* is the speed of water (m/s) and *A* is the area perpendicular to the direction of water flow. Five values of speed of water are considered: 1, 1.5, 2, 3, 5 m/s. Through a sensitivity analysis, a better modal adjustment was identified for values of speed of water of 3 m/s and 2 m/s for AVTs and VITs, respectively. In Figure 16 and Figure 17, the values of the MAC index for the updated FE model are shown with respect to the initial values, showing improvements up to 5% on average. In modes with frequencies lower than 1.8 Hz, the improvement is around 0.5%, while for higher frequencies a 20% modal improvement is reached. With respect to the percentage of error in frequency, this parameter decreases, on average, up to 1.5%.



(1)
$$\mathrm{AASHTO}\;\;\;m=\frac{\left(C_{D}\frac{w}{2g}V^{2}\right)\cdot A}{g}\;\left(\frac{\mathrm{kg}}{g}\right)$$



### 6.2. The Incidence of Sensor Location in AVTs with Full Time-Series Length

Figure 18 shows the average values of MAC indices for AVT1 (L/10), AVT2 (L/6), AVT3 (L/6), and AVT4 (L/10) considering full time-series length. As previously mentioned, the objective was one AVT per day, and therefore time-series length defined forAVT1 and AVT4 is 20 min in contrast with the 35 min selected for AVT2 and AVT3. It is possible to observe that the higher value of the MAC index is associated with AVT2 (L/6), with values similar to OSP configurations AVT3 (L/6) and AVT4 (L/10). Figure 18 shows the percentage of relative errors between the MAC index of AVT2 (L/6) and OSP configurations AVT3 (L/6) and AVT4 (L/10) with respect to AVT1 (L/10), as well as the percentage of relative error between the MAC index of AVT2 (L/6) and OSP configuration AVT3 (L/6) with respect to AVT4 (L/10). The sensor-to-sensor separation distance of L/6 shows improved results, suggesting a marked incidence of sensor position rather than time-series length. Regarding the hybrid configurations, the distribution AVT4 (L/10) shows improvements in modal identification of 9% with respect to AVT1 (L/10), but there is not a significant difference between the results of OSP configurations AVT3 (L/6) and AVT3 (L/10).

### 6.3. The Incidence of Time-Series Length in AVTs and Adjusted Time-Series Length to 20 min

The time-series length of AVT2 and AVT3 is reduced to 20 min. The average values of MAC indices for AVT1 (L/10), AVT2 (L/6), AVT3 (L/6), and AVT4 (L/10) are shown in Figure 19. The percentage of relative errors of the MAC indices for AVT2 and AVT3 (L/6 configurations) are computed with respect to AVT1 and AVT4 (L/10 configurations), as shown in Figure 19. The results show a similar tendency found in the previous parameter. The sensor-to-sensor separation distance dominates over the time-series length since, in L/6 configurations, the modal consistency shows slight variations of approximately 2%. The time-series length is decisive in mode shapes that are not very excited in terms of frequency and magnitude, such as the vertical and longitudinal modes, as shown in Table 3 for the 1-Z, 3-Z, and 3-X modes.

### 6.4. The Incidence of Excitation Source between AVTs and VITs with Full Time-Series Length

Figure 20a shows the average values of MAC indices for all the field tests performed in this research work. Figure 20b shows the relative percentages of error in the AVTs concerning the VITs configurations by considering that it is only possible to relate test results that have been taken with the same relative sensor separation (L/6 and L/10). It is important to highlight that AVT1 and AVT4 were conducted using a time-series length of 20 min, AVT2 and AVT3 using a time-series length of 35 min and VIT1 and VIT2 using a time-series length of 10 min. It is possible to observe that improvements of up to 12% in L/6 configurations are achieved with respect to AVTs in contrast with L/10 configurations. Figure 20c shows that the modal shapes 2-Y, 3-Y, and 1-Z are associated with the highest MAC index increment when compared to that obtained in AVTs, with 1-Z being the best-identified mode shape, in contrast with the 1-Y mode shape, which is widely affected by the reduction in the time-series length. The improvement in dynamic identification with respect to AVTs is 2% for the 11 modal shapes of interest. Although there is a not significant difference, correct results are obtained in half the test time. Forces generated by the truck improve vertical mode identification by increasing modal consistency.

### 6.5. The Incidence of OSP in VITs with Respect to AVTs with Full Time-Series Length

Figure 21 shows the results of modal consistency for OSP configurations by considering the two excitation sources used in this investigation. The improvement in results, on average, exceeds 4%. These results allow us to conclude that with few OSP locations, modal identification is optimal and even improves in VITs, which generally represents a reduction in execution time and therefore operating costs.

### 6.6. The Incidence of AVTs with Time-Series Length Equals to VITs’ Time-Series Length (10 min)

Finally, the time-series length is set to 10 min based on the time-series length of VITs to analyze the incidence of a considerably reduced time-series length in modal identification. Figure 22 shows the MAC index values for all the field tests performed in this research work. There are reductions in modal identification results on average of 3.5%. The greatest affectations are in vertical mode shapes with reductions in modal identification results of up to 15%. However, the transverse mode shapes show improvements in modal identification of 15%. This situation again suggests that the time-series length is relevant in slightly excited modal shapes with respect to the dominant direction of the excitation force. The primary economic constraint in field testing is related to limited sensor availability. In addition, long-span PC bridges have suffered from incremental deflection, and the correct identification of vertical mode shapes is crucial to update FE models to study long-term deflection and improve current design standards allowing the construction of longer bridge spans. It is also important to note that environmental factors play a central role in modal identification. In this framework, the time-series length used in field testing must be carefully selected to minimize the influence of such parameters. This research work aimed to conduct one test per day during daylight hours to minimize such environmental effects. Therefore, the most influential factor in time-series length is sensor availability, which is mainly affected by economic constraints. On the other hand, the use of a vehicle-induced load to excite the bridge implies the closure of the bridge to traffic, and time-series length also becomes relevant.

### 6.7. Cost Comparison

Optimizing the economic resources available to execute dynamic tests is an economic challenge, especially in selecting the excitation source. Therefore, the use of trucks or mechanical devices is not a profitable option in most dynamic tests on bridges, not only regarding the cost of these devices, but also the costs associated with closing the bridge to traffic. Table 4 shows the representative costs of the dynamic tests executed in the present study based on three representative items, signalling, transport, and support staff. The cost per test day is six times higher in FVTs than in AVTs. The logistical complexity of VITs justifies the increase. In the first instance, when analyzing the modal improvement with respect to AVT (Figure 20), the use of additional forces in dynamic tests is not a profitable investment if it is used in a conventional instrumentation mesh (L/6-18 measurement points or more). However, if only the OSP locations are instrumented, keeping 35 min in AVT and only 10 min in VIT, the monetary difference is reduced to 2.26 times, as shown in Table 5. This analysis shows that there is a clear advantage in dynamic tests using optimally selected locations. At this point, it is economical to conduct VIT tests to improve the identification of vertical modes, especially if the objective is to predict long-term deflections, given the complex nature of long-term deflections of long-span PC bridges. It is important to note that the signalling equipment is reusable, so the financial ratio between tests can decrease by between 10% and 20%.

## 7. Conclusions

An adequate preliminary analysis based on FE models is essential to identify the most critical parameters to be considered in field tests. This type of preliminary analysis should be approached with a detailed study of previous experiences carried out in bridges with similar characteristics. The next step is to build numerical models that allow the study of relevant variables that can affect the tests that are planned to be executed in the field. In this regard, it is essential to be clear about the economic restrictions related to field testing, such as sensor availability, the effect of bridge daylight closures, environmental factors, and that the tests can be carried out preferably in daylight to minimize the influence of environmental factors and accidents, especially in tests carried out with trucks. Finally, during field testing, special attention must be given to data collection of operational and environmental factors. Based on the previously described methodology, a three-dimensional FE model was established for the Tablazo bridge using construction drawings. Subsequently, a detailed review of the existing literature related to tests carried out on similar bridges made it possible to list a set of critical parameters to consider in field tests, such as main span length (L), excitation source, sensor-to-sensor separation distance, time-series length, and sampling frequency. Then, numerical simulations were carried out considering the number of available sensors and suitable configurations for vehicle loading to identify improvements in modal identification using OSP configurations. It was found that modal identification in the numerical simulations not only depended on the load case, but also on the sensor locations, obtaining better results in distributions with remarkable similarity or closeness to OSP configurations.

The selection of the mode shapes of interest for the calculation of the OSP configurations depends to a great extent on field testing variables, and therefore operational and environmental factors were collected during field testing. A set of optimal sensor positions was identified that allowed the identification of mode shapes with great consistency and the least investment of total test time. A relationship was found between the optimal sensor distribution in FE modelling and field testing. This suggests that carrying out a previous OSP estimation based on the experience of the possible modal configurations to be identified would allow a more accurate distribution of sensors. Finally, modal identification depends to a greater extent on the location of the sensor rather than on the time-series length by location. Time-series length is decisive in low-excited modes. Although VITs provide improvements of between 4% and 12% of the dynamic results of AVTs, the cost of execution using vehicles is 2–6 times higher than in AVTs, suggesting that these VITs can be used to improve modal identification of vertical mode shapes when ambient loading poorly excites such mode shapes. The vehicle load proposal used in this research work enhances the identification of vertical mode shapes with respect to the AVTs results. Environmental factors, specifically fluid–structure interaction, actively participate in the dynamic response of the bridge and therefore in the present study are considered as the first factor to be updated in the FE model by adjusting the mass and stiffness of the structure before using the FE model in the prediction of long-term deflections of long-span PC bridges.

## Figures and Tables

**Figure 1 sensors-22-01219-f001:**
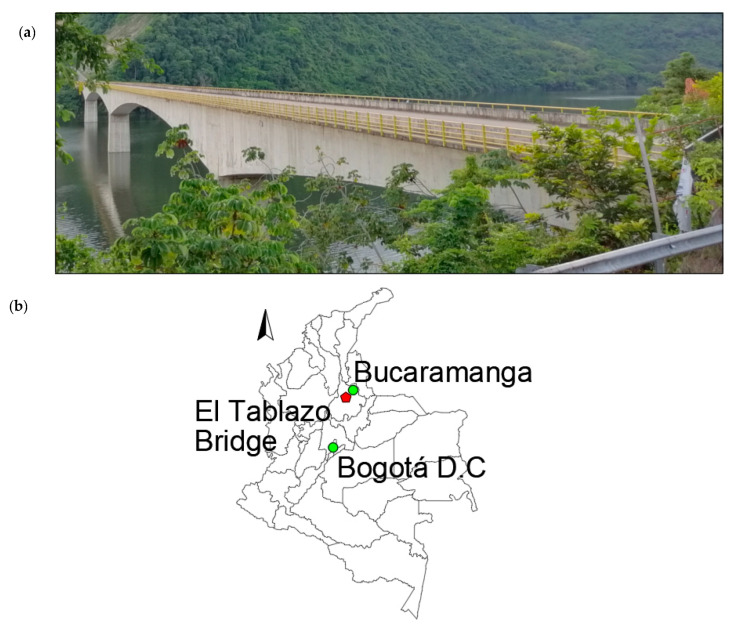
(**a**) The Tablazo bridge view from south access, (**b**) Location of the bridge.

**Figure 2 sensors-22-01219-f002:**
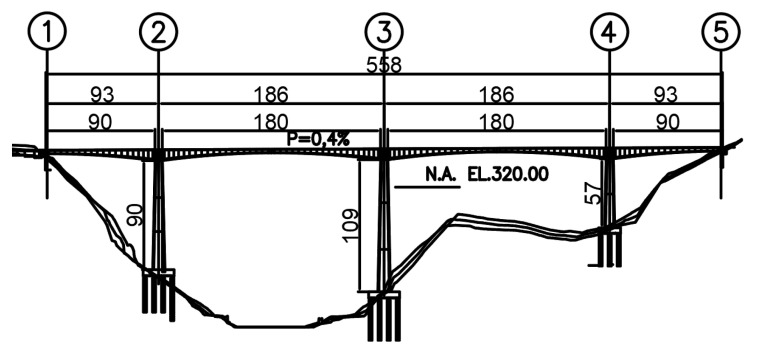
The Tablazo bridge side view.

**Figure 3 sensors-22-01219-f003:**
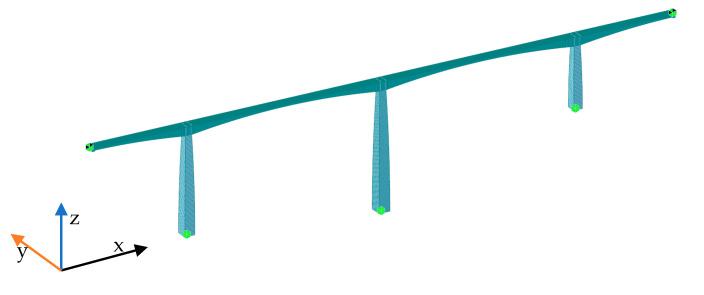
FE model of the Tablazo bridge MIDAS Civil©.

**Figure 4 sensors-22-01219-f004:**
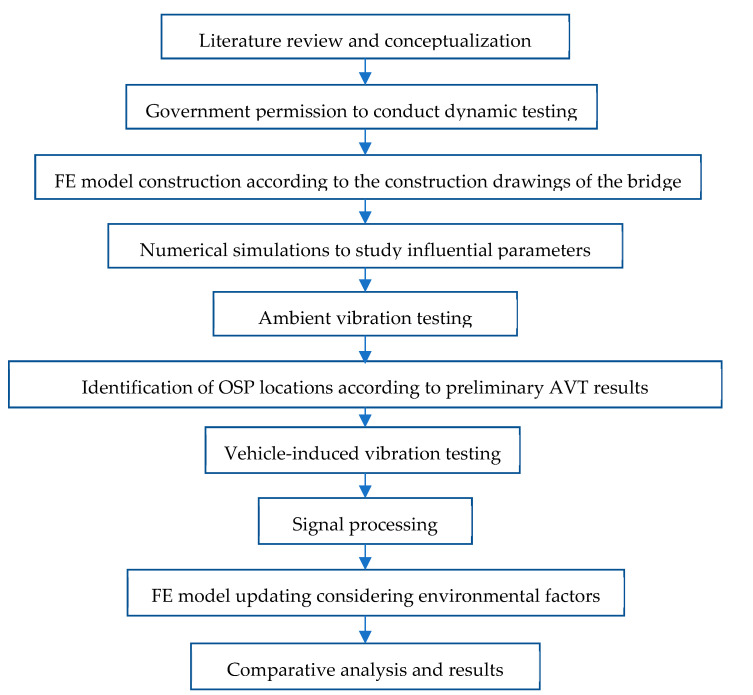
Schematic flow chart showing field study.

**Figure 5 sensors-22-01219-f005:**
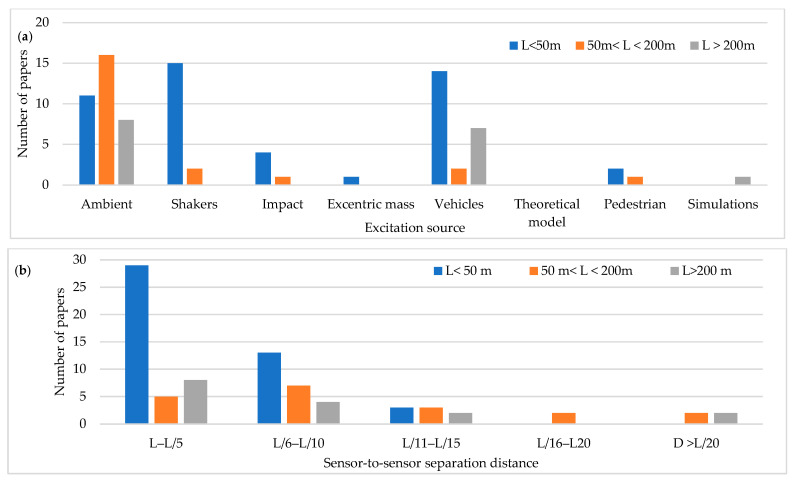
Distribution of the variables as a function of the main span length L: (**a**) Excitation source (**b**) sensor-to-sensor separation distance (**c**) Sampling frequency (**d**) Time-series length.

**Figure 6 sensors-22-01219-f006:**
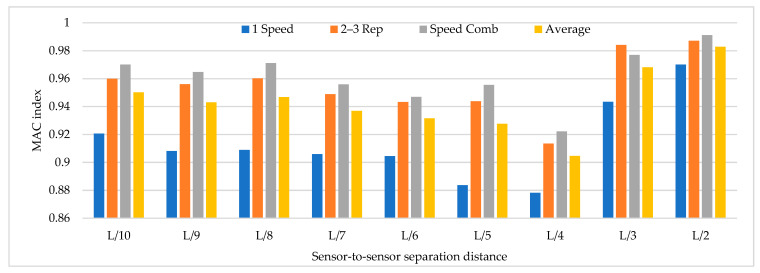
MAC index results for different sensor-to-sensor separation distances.

**Figure 7 sensors-22-01219-f007:**
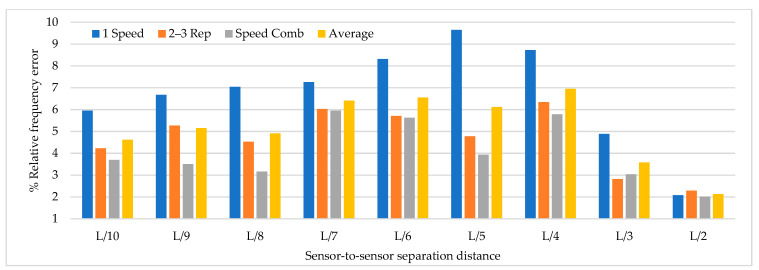
Percentage of relative frequency error for different sensor-to-sensor separation distances.

**Figure 8 sensors-22-01219-f008:**
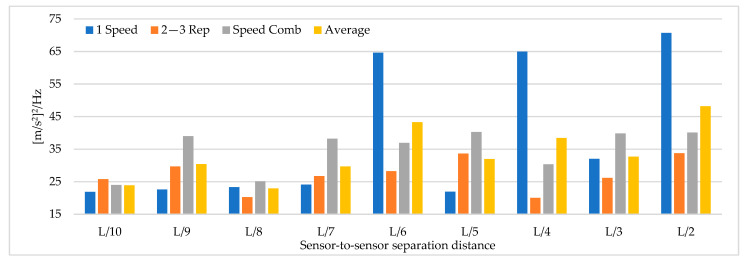
The magnitude of the power spectrum for different sensor-to-sensor separation distances (Units: [m/s^2^]^2^/Hz).

**Figure 9 sensors-22-01219-f009:**
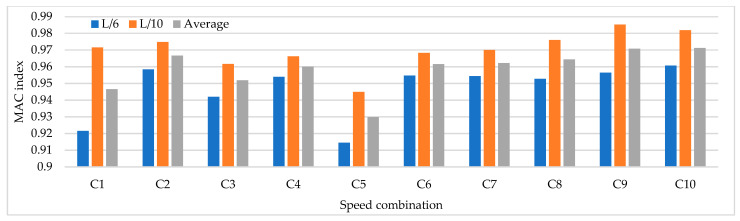
MAC index results for speed combinations for sensor-to-sensor separation distances of L/6 and L/10.

**Figure 10 sensors-22-01219-f010:**
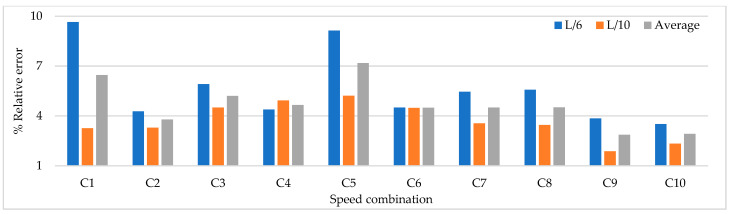
Percentage of relative frequency error for different speed combinations (L/6 and L/10).

**Figure 11 sensors-22-01219-f011:**
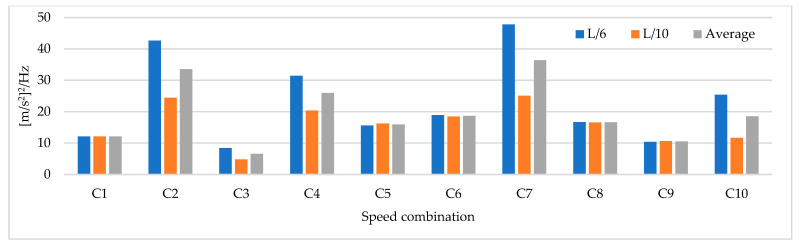
Magnitude of the power spectrum for different speed combinations (L/6 and L/10) (Units: [m/s^2^]^2^/Hz).

**Figure 12 sensors-22-01219-f012:**
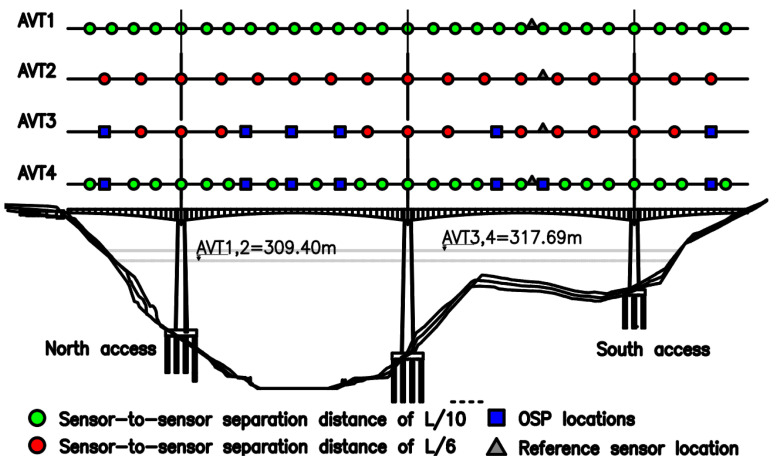
Location of sensors in the four days of tests and the level of water measured in the field.

**Figure 13 sensors-22-01219-f013:**
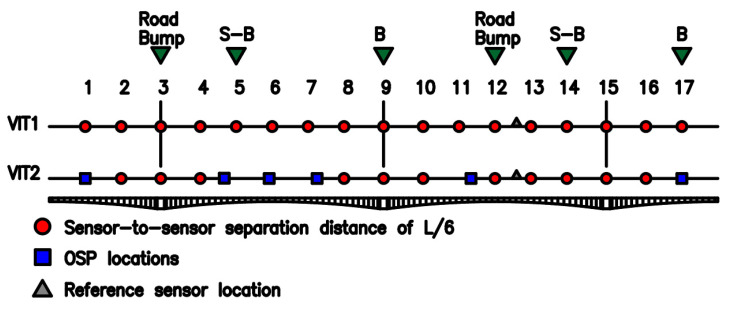
Location of sensors in VITs and braking B (moving forward) and S-B (moving in reverse).

**Figure 14 sensors-22-01219-f014:**
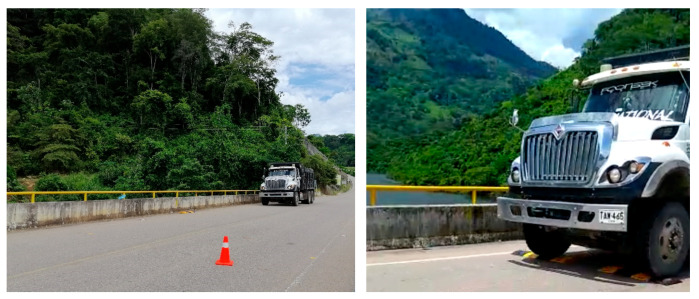
Three-axle 32-ton truck moving at a speed of 40 km/h on the Tablazo bridge.

**Figure 15 sensors-22-01219-f015:**
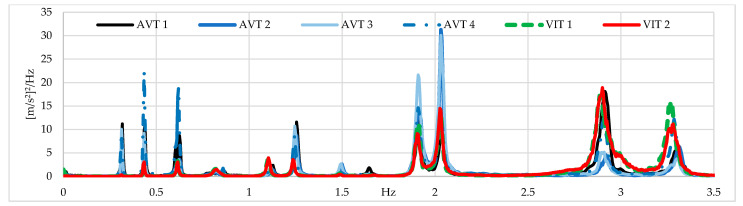
The power spectrum for AVTs and VITs (Units: [m/s^2^]^2^/Hz).

**Figure 16 sensors-22-01219-f016:**
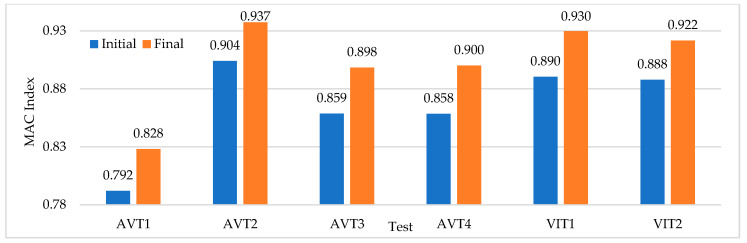
Variation of the MAC index due to update of the FE model.

**Figure 17 sensors-22-01219-f017:**
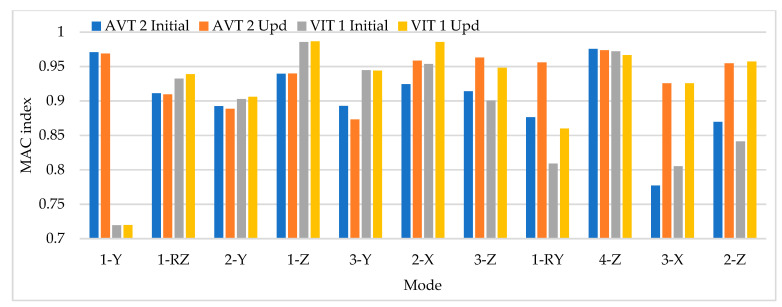
MAC index update by identified experimental modes.

**Figure 18 sensors-22-01219-f018:**
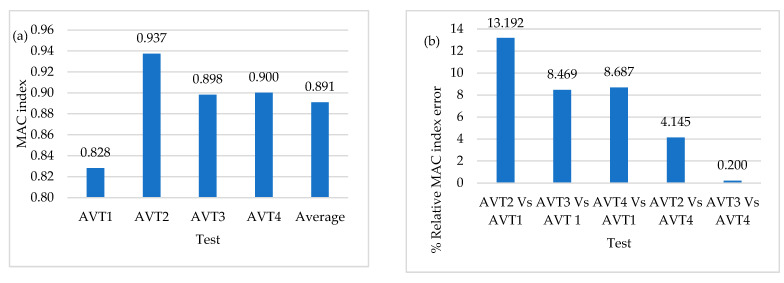
(**a**) Average values of MAC index of AVTs (**b**) Percentage of relative MAC index error of AVTs with full time-series length.

**Figure 19 sensors-22-01219-f019:**
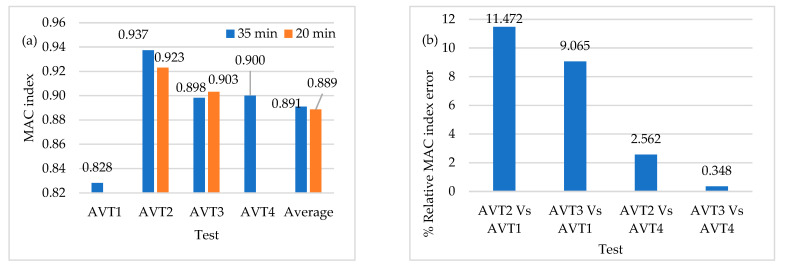
(**a**) Average values of MAC index (**b**) Percentage of relative MAC index error of AVTs with reduced time-series length.

**Figure 20 sensors-22-01219-f020:**
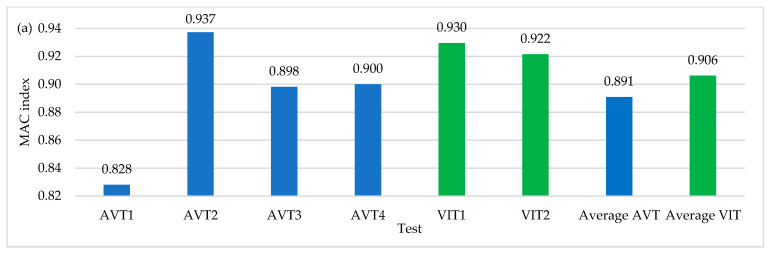
(**a**) Average values of MAC index (**b**) Percentage of relative MAC index error of excitation sources (**c**) Mode shape MAC index variation with respect to AVTs.

**Figure 21 sensors-22-01219-f021:**
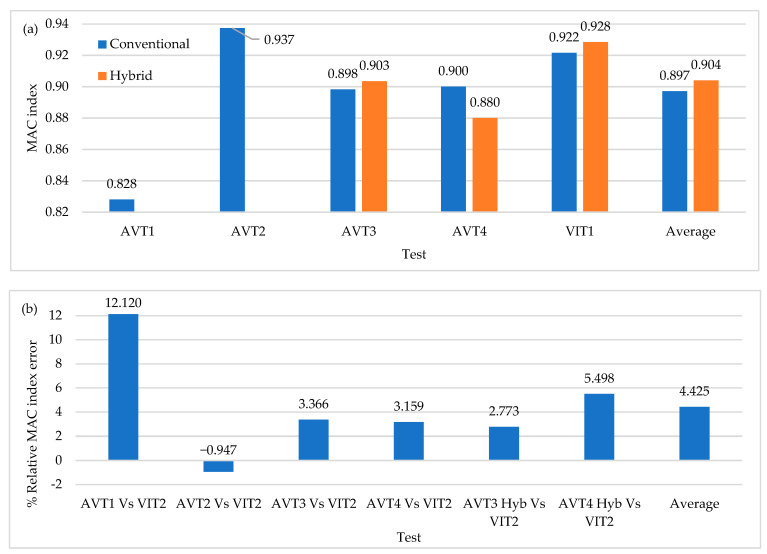
(**a**) Average values of MAC index (**b**) Percentage of relative MAC index error of OSP locations.

**Figure 22 sensors-22-01219-f022:**
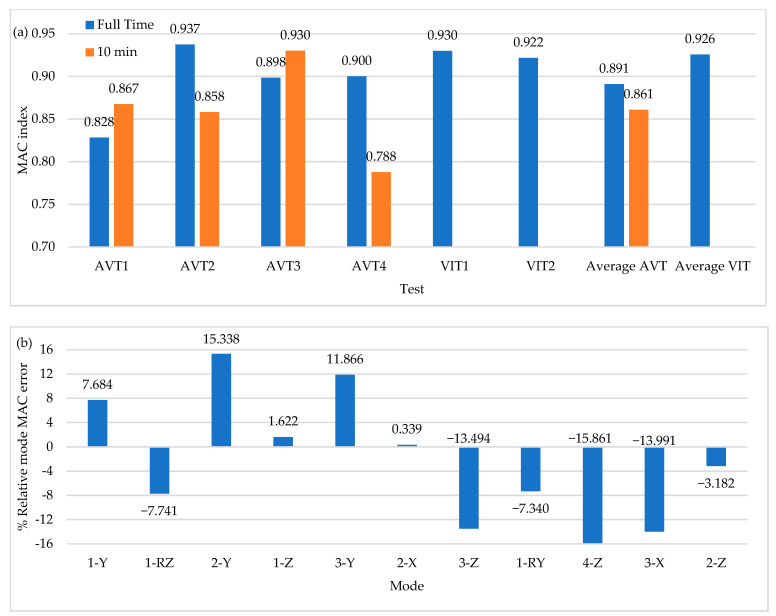
(**a**) Average values of MAC index (**b**) Percentage of relative MAC index error of AVTs with 10 min time series.

**Table 1 sensors-22-01219-t001:** Frequencies and modal participation factors obtained from the FE model.

Mode	Freq.	Tran.-X	Tran.-Y	Tran.-Z	Rotn.-X	Rotn.-Y	Rotn.-Z
1-Y	0.345	0	47.22	0	15.25	0	0.13
1-X	0.492	64.85	0	0	0	0	0
1-RZ	0.477	0	0.02	0	0	0	32.9
2-Y	0.685	0	11.45	0	7.41	0	0.11
1-Z	0.841	0.56	0	12.49	0	0.8	0
3-Y	1.281	0	10.62	0	0.15	0	0.75
2-X	1.36	8.57	0	1.1	0	1.55	0
3-Z	1.48	1.09	0	7.37	0	0.06	0
1-RY	1.742	0.06	0	0	0	21.7	0
4-Z	1.942	0	0	3.28	0	0.02	0
3-X	3.032	2.74	0	0.92	0	1.4	0
2-Z	3.117	0	0	9.52	0	0.64	0

**Table 2 sensors-22-01219-t002:** Average values of MAC indices of the 11 identified mode shapes.

	AVT				VIT		AVT OSP		VIT
Mode	T1	T2	T3	T4	T1	T2	T2	T3	OSP T2
1-Y	0.76	0.97	0.71	0.94	0.77	0.72	0.78	0.95	0.67
1-RZ	0.89	0.91	0.94	0.81	0.91	0.94	0.72	0.79	0.99
2-Y	0.57	0.89	0.90	0.76	0.92	0.91	0.83	0.85	0.92
1-Z	0.62	0.94	0.82	0.92	0.99	0.99	0.93	0.90	0.99
3-Y	0.78	0.87	0.70	0.89	0.93	0.94	0.92	0.92	0.95
2-X	0.96	0.96	0.98	0.97	0.99	0.99	0.99	0.98	0.99
3-Z	0.75	0.96	0.98	0.83	0.96	0.95	0.98	0.82	0.94
1-RY	0.95	0.96	0.97	0.96	0.93	0.86	0.97	0.96	0.92
3-Z	0.96	0.97	0.99	0.96	0.98	0.97	0.99	0.96	0.99
3-X	0.95	0.93	0.92	0.92	0.89	0.93	0.85	0.73	0.93
2-Z	0.93	0.95	0.98	0.93	0.95	0.96	0.97	0.83	0.94

**Table 3 sensors-22-01219-t003:** Experimental frequencies obtained from dynamic tests.

	AVT1			AVT2			AVT3			AVT4			VIT1			VIT2		
Mode	FDD	EFDD	SSI	FDD	EFDD	SSI	FDD	EFDD	SSI	FDD	EFDD	SSI	FDD	EFDD	SSI	FDD	EFDD	SSI
1-Y	0.32	0.32	0.36	0.33	0.32	0.36	0.32	0.31	0.34	0.33	0.31	0.38	0.31	0.31	0.40	0.32	0.31	0.45
1-RZ	0.44	0.44	0.46	0.42	0.44	0.47	0.48	0.44	0.45	0.45	0.43	0.45	0.44	0.43	0.49	0.43	0.43	0.45
2-Y	0.62	0.63	0.65	0.64	0.63	0.62	0.63	0.62	0.62	0.62	0.62	0.62	0.62	0.62	0.61	0.62	0.61	0.63
1-Z	0.85	0.86	0.81	0.79	0.82	0.81	0.81	0.82	0.82	0.80	0.81	0.82	0.84	0.82	0.82	0.81	0.82	0.82
3-Y	1.12	1.13	1.10	1.12	1.13	1.12	1.12	1.12	1.12	1.12	1.11	1.15	1.10	1.10	1.10	1.05	1.10	1.12
2-X	1.26	1.26	1.25	1.23	1.25	1.26	1.22	1.25	1.25	1.23	1.24	1.26	1.20	1.24	1.24	1.19	1.24	1.27
3-Z	1.49	1.50	1.50	1.45	1.48	1.49	1.48	1.48	1.50	1.49	1.49	1.53	1.48	1.49	1.49	1.48	1.49	1.49
1-RY	1.87	1.93	1.91	1.88	1.97	1.91	1.92	1.96	1.91	1.91	1.94	1.91	1.90	1.95	1.90	1.89	1.96	1.90
4-Z	2.03	2.00	2.03	2.03	2.02	2.04	2.06	2.02	2.03	2.04	2.01	2.03	2.03	2.00	2.03	2.02	2.02	2.03
3-X	2.93	2.92	2.92	2.97	2.92	2.92	2.88	2.91	2.91	2.91	2.90	2.91	2.85	2.90	2.90	2.94	2.90	2.90
2-Z	3.35	3.29	3.31	3.35	3.30	3.31	3.26	3.29	3.26	3.34	3.28	3.28	3.32	3.27	3.27	3.30	3.27	3.28

**Table 4 sensors-22-01219-t004:** Representative cost of dynamic tests per day (8 h).

Item	AVT [u]	VIT [u]
Signalling	1.4	14
Transport	7	45
Support staff	2.9	9
Total	11.3	68
VIT/AVT	6.0	

**Table 5 sensors-22-01219-t005:** Representative cost of dynamic tests using OSP configurations.

	Cost per Hour [u]		Total OSP Cost [u]	
Item	AVT	VIT	AVT = 5.33 h	VIT = 2 h
Signaling	0.175	1.75	0.93	3.5
Transport	0.875	5.625	4.67	11.25
Support staff	0.3625	1.125	1.93	2.25
Total	1.4125	8.5	7.53	17
VIT/AVT	6.02		2.26	

## Data Availability

The data that support the findings of this study are available from the corresponding author, upon reasonable request.

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
