# Peer review of "Vehicle Bump Testing Parameters Influencing Modal Identification of Long-Span Segmental Prestressed Concrete Bridges"

_sensors, 2022, doi:10.3390/s22031219_

Round 1

Reviewer 1 Report

This paper presents the results of field measurements at optimally selected locations using vehicle‐induced excitations consisting of a 32‐ton truck and a springboard with a height of 50 mm. There are some issues that should be well considered in the manuscript.

  1. In the first part "Introduction", the authors should in troduce more conten about the progress of modal parameter identification.
  2. In the part of "5. Comparisons", the modal parameters are identified by the commercial software "ARTeMIS'. However, some key issues are unkown. For example, what methods are used to identify the modal parameters? How the set the parameters of the method? The authors must give a clear description of the identification progress.

Author Response

1. “In the first part “introduction”, the authors should introduce more content about the progress of modal parameter identification”

Based on a group of references provided by the other reviewer more content related to modal identification was included in the introduction from lines 33 to 54. Reference 5 (recommended by the other reviewer) provided information from measurements collected in a prestress bridge during 14 years.

“In the part of “5. Comparisons”, the modal parameters are identified by the commercial software “ARTeMIS”. However, some key issues are unknown. For example, what methods are used to identify the modal parameters? How the set the parameters of the method? The authors must give a clear description of the identification progress.

A comparative analysis is included in the paper in the new section (FE model updating). Based on a comment from the other reviewer we included two new sections in the paper, the first section named "FE model updating" and the second section named "Cost comparison".  We included the requested information in both numerical simulations and field testing, we also added a new table showing modal identification results from different methodologies and in the section FE model updating we included two new figures (Figures 14 and 15) showing the results from the FE model updating.

The abstract was also modified to show the main research outputs. A new figure (Figure 4) was included to show the flow chart of the field study.  Fluid-structure interaction was included in the FE model updating.

Finally, the whole document was reviewed for potential error using a software developed to detect common grammatical mistakes 

Reviewer 2 Report

The paper demonstrates the comparison of ambient vibration tests and vehicle indued tests on an existing long-span prestressed concrete bridge to identify its modal frequencies and mode shapes. Apart from comparing the ambient vibration test results with 32-ton truck vehicle-induced vibration tests, the study also includes the effects of variations in sensor locations and length of acquired accelerometer time history. The paper addresses the important study of field tests looking at various factors that are relevant for structural health monitoring of bridges. However, to bring out the primary contribution and improve the organizational flow of the paper, the authors are suggested to address the following comments and modify the paper accordingly.

  1. In the introduction section, the authors need to explicitly mention the knowledge gap that remains and how the present paper contributes to filling in some of those gaps.
  2. To improve the flow of information, the authors are suggested to include a figure at the beginning that will inform the readers about the methodology that is followed in the field study, from conceptualization using the FEM model to final processing of acquired data.
  3. The authors are suggested to enumerate the key inferences of the comparative field study either in the discussion or in the conclusion section.
  4. Before performing the field tests, the results from the FEM simulation of the bridge are used for efficient planning. After the field tests, the authors report the identified modal parameters. In this context, the authors are suggested to report the comparison of the modal parameters from the preliminary FEM model and the ones obtained through field tests. Also, the authors need to comment on their differences and how the field-test results can be used to update the FEM model.
  5. The authors are suggested to briefly describe the method used for obtaining the optimal sensor placements.
  6. Also, the authors need to mention which modal analysis technique is used to identify the modal parameters both for the field-test and the numerical simulations. Do the authors think the results may vary if any other modal analysis technique is used?
  7. Several environmental factors affect the identified modal parameters such as temperature, humidity to name a few. The authors mention environmental factors both in the introduction and in the results section. However, it is not clear from the manuscript which variable environmental factors were measured in the field study and what kind of related variations were observed in the identified modal parameters.
  8. In the abstract, the authors mention about the cost ratio of forced vibration test and ambient vibration test is reduced to 2. However, it is not clear from the manuscript how is the cost of those tests evaluated and compared with the previously reported cost, the significance of such reduction in cost ratio, or how does that relate to the main contribution of the paper.
  9. The authors are suggested to add these references to improve the literature review section
  • Ko, J. M., & Ni, Y. Q. (2005). Technology developments in structural health monitoring of large-scale bridges. Engineering structures27(12), 1715-1725.
  • Catbas, F. N., Susoy, M., & Frangopol, D. M. (2008). Structural health monitoring and reliability estimation: Long span truss bridge application with environmental monitoring data. Engineering Structures30(9), 2347-2359.
  • Sun, L., Shang, Z., Xia, Y., Bhowmick, S., & Nagarajaiah, S. (2020). Review of bridge structural health monitoring aided by big data and artificial intelligence: From condition assessment to damage detection. Journal of Structural Engineering146(5), 04020073.
  • Hu, W. H., Tang, D. H., Teng, J., Said, S., & Rohrmann, R. (2018). Structural health monitoring of a prestressed concrete bridge based on statistical pattern recognition of continuous dynamic measurements over 14 years. Sensors18(12), 4117.
  • Xu, Y., Brownjohn, J. M., & Huseynov, F. (2019). Accurate deformation monitoring on bridge structures using a cost-effective sensing system combined with a camera and accelerometers: Case study. Journal of Bridge Engineering24(1), 05018014.

Additional minor comments,

  1. Line 147, the authors need to briefly describe and provide a reference for the OSP method “Effective Independence”.
  2. Line 169, the authors need to briefly describe the meaning of “dynamic tremor alpha=0”.
  3. Line 233, the authors need to add references for the statement “It was found in the literature review…widely used in field testing”.
  4. Line 249, modal properties of the FEM model corresponding to the mode 1-X(f=0.43 Hz) are not listed in Table 1.
  5. Line 302, the authors need to elaborate on the significance of the reported water levels.
  6. Throughout the text, the authors mention forced vibration tests (FVTs) as well as vehicle-induced tests (VITs). If they are different, the authors need to explicitly mention the differences, else the authors need to follow a consistent naming convention.

Author Response

1. "In the introduction, the authors need to explicitly mention the knowledge gap that remains and how the present paper contributes to filling in some of those gaps"

We included new explanations in the introduction (lines 162 to 166, 169 to 170, 175 to 176). Based on the references kindly provided by you (references 2 and 5) we analyze the temperature effect in modal identification (lines 34 to 54). We identified 2 main gaps. The first is current code limitations and temperature effects in modal identification. The situation in Colombia is that long-span PC bridges are becoming widely adopted across the country. The current bridge design code does not provide guidelines for dynamic testing. The main objective is to propose a one-day test. The two main reasons are minimizing the temperature effect on modal identification and optimizing AVT and VIT to be executed in one day to reduce costs. We included a new section in the paper (cost comparisons) to show the advantages of the proposed methodology.

2. "To improve the flow of information, the authors are suggested to include a figure at the beginning that will inform the readers about the methodology that followed in the field study, from conceptualization using the FEM model to final processing of acquired data."

The suggested figure was included (Figure 4).

3. "The authors are suggested to enumerate the key inferences of the comparative field study either in the discussion or in the conclusion section"

The key inferences were enumerated (lines 422 to 430).

4. "Before performing the field test, the results from the FEM simulation of the bridge are used for efficient planning. After the field test, the authors are suggested to report the comparison of the modal parameters from the preliminary FEM model and the ones obtained through field tests. Also, the authors need to comment on their differences and how the field-test results can be used to update the FEM model"

We included a new section named FE model updating; we conducted FE model updating by considering water-structure interaction and comparing initial FE results and results from the updated FE model to provide a more robust analysis. We decided to perform FE updating based on your comment related to a deeper analysis of environmental factors; we consider that water-structure interaction is relevant in this study as an environmental factor and temperature effects. We included a new section and two new figures (Figures 13 and 14).

5. "The authors are suggested to briefly describe the method used for obtaining the optimal sensor placement"

The description was included in lines 364 to 372.

6. "Also, the authors need to mention which modal analysis technique is used to identify the modal parameters both for the field test and the numerical simulations. Do the authors think the results may vary if any other modal analysis technique is used?"

It was included the explanation for the numerical simulations in lines 312 to 317. We considered that the reviewer comment is relevant, and therefore, we added a new table (Table 2) to compare results from three different modal identification techniques before conducting FE model updating to support the incidence of environmental factors (lines 432 to 453).

7. "Several environmental factors affect the identified modal parameters such as temperature, humidity to name a few. The authors mention environmental factors both in the introduction and the results section. however, it is not clear from the manuscript which variable environmental factors were measured in the field study and what kind of related variations were observed in the identified modal parameters"

Your comment greatly helped us to contextualize the importance of environmental factors. Therefore, we conducted new simulations in the context of the FE model updating to consider water-structure interaction. In our opinion, the most relevant environmental factor in this study. We also are grateful for providing us with reference 2, which helped us to better contextualize that our main objective from the beginning was to propose a one-day test to minimize the effect of temperature and reduce costs. We included in the paper the section named "6.7 FE model updating", where we analyzed the effect of water-structure interaction to provide an updated model as an output of this research work.

8. "In the abstract, the authors mentions about the cost ratio of forced vibration test and ambient vibration test is reduced to 2. however, it is not clear from the manuscript how is the cost of those test evaluated and compared with the previously reported cost, the significance of such reduction in cost ratio, or how does that relate to the main contribution of the paper"

Thanks to your comment, we contextualize this vital research output in a new section named "5.6 cost comparisons" to support the cost ratio based on the data provided by two new tables (Tables 4 and 5). We also provided a detailed explanation in lines 556 to 573. 

9. "The authors are suggested to add these references to improve the literature review section"

The suggested references were added [2]-[6]. 

10. Line 147, the authors need to briefly describe and provide a reference for the OSP method "Effective independence"

The reference was provided: 

[36] D. C. Kammer, "Sensor Placement for On-Orbit Modal Identification and Correlation of Large Space Structures," Journal of Guidance, Control, and Dynamics, vol. 14, no. 2, pp. 251–259, 1991, doi: https://doi.org/10.2514/3.20635

11. “Line 169, the authors need to briefly describe the meaning of dynamic tremor alpha=0”

(dynamic tremor alpha=0, which represents a vertical action proportional to the value of the vehicle weight) (lines 192 to 193).

12. "Line, 233, the authors need to add references for the statement "it was found in the literature review... widely used in field testing"

The references were added

It was found in the literature review that trucks weighing more than 20 tons and travelling at speeds of between 10 and 80 km/h are widely used in field testing [10], [27], [30]-[34]. (line 292).

13. "Line 249, modal properties of the FEM model corresponding to the mode 1-X f= 0.43 are not listed in table 1"

The correction was made in line 213 Table 1.

14. "Line 302, the authors need to elaborate on the significance of the reported water levels"

The significance of the water levels was included and connected to the FE model updating section, a new section in the paper based on the reviewer comment.

15. "Throughout the text, the authors mention forced vibration tests (FVTs) as well as vehicle-induced tests (VITs). If they are different. The authors need to explicitly mention the differences, else the authors need to follow a consistent naming convention"

It was corrected FVT in the abstract and in line 84 the explanation requested by the reviewer is included

However, such deployments are currently restricted to small- and medium-span bridge structures. Due to limitations in estimating the excitation force exerted by a moving vehicle, VIT is not classified as a FVT.